# Galectin-1 Modulates the Fusogenic Activity of Placental Endogenous Retroviral Envelopes

**DOI:** 10.3390/v15122441

**Published:** 2023-12-16

**Authors:** Caroline Toudic, Maike Maurer, Guillaume St-Pierre, Yong Xiao, Norbert Bannert, Julie Lafond, Éric Rassart, Sachiko Sato, Benoit Barbeau

**Affiliations:** 1Département des Sciences Biologiques and Centre d’excellence en Recherche sur les Maladies Orphelines-Fondation Courtois, Université du Québec à Montréal, Montréal, QC H3C 3P8, Canada; caroline.toudic22@gmail.com (C.T.); xiao.yong@uqam.ca (Y.X.); lafond.julie@uqam.ca (J.L.); rassart.eric@uqam.ca (É.R.); 2Robert-Koch Institute, 13353 Berlin, Germany; m.maurer94@gmx.de (M.M.); bannertn@rki.de (N.B.); 3Glycobiology and Bioimaging Laboratory, Research Centre for Infectious Diseases and Axe Maladies Infectieuses et Immunitaires, Laval University, Quebec City, QC G1V 0A6, Canada; guillaume.st-pierre@crchudequebec.ulaval.ca (G.S.-P.); sachiko.sato@crchudequebec.ulaval.ca (S.S.); 4Regroupement Intersectoriel de Recherche en Santé de l’Université du Québec, Montréal, QC H2X 1E3, Canada

**Keywords:** human endogenous retrovirus, Syncytin-1 and -2, EnvP(b), galectin-1, cell fusion

## Abstract

Syncytin-1 and -2 are glycoproteins encoded by human endogenous retrovirus (hERV) that, through their fusogenic properties, are needed for the formation of the placental syncytiotrophoblast. Previous studies suggested that these proteins, in addition to the EnvP(b) envelope protein, are also involved in other cell fusion events. Since galectin-1 is a β-galactoside-binding protein associated with cytotrophoblast fusion during placental development, we previously tested its effect on Syncytin-mediated cell fusion and showed that this protein differently modulates the fusogenic potential of Syncytin-1 and -2. Herein, we were interested in comparing the impact of galectin-1 on hERV envelope proteins in different cellular contexts. Using a syncytium assay, we first demonstrated that galectin-1 increased the fusion of Syncytin-2- and EnvP(b)-expressing cells. We then tested the infectivity of Syncytin-1 and -2 vs. VSV-G-pseudotyped viruses toward Cos-7 and various human cell lines. In the presence of galectin-1, infection of Syncytin-2-pseudotyped viruses augmented for all cell lines. In contrast, the impact of galectin-1 on the infectivity of Syncytin-1-pseudotyped viruses varied, being cell- and dose-dependent. In this study, we report the functional associations between three hERV envelope proteins and galectin-1, which should provide information on the fusogenic activity of these proteins in the placenta and other biological and pathological processes.

## 1. Introduction

Human endogenous retrovirus (hERV) have repeatedly and independently integrated the human genome during evolution and currently constitute around 8% of our DNA [1,2]. Most hERV sequences have been silenced by mutations or deletions, but a handful of hERV *env* genes have been selectively conserved and are still transcriptionally active in different tissues [3]. Among them, *ERVWE1* and *ERVFRD1* have been the focus of many studies, as they respectively code for Syncytin (Syn)-1 and -2, two envelope glycoproteins that are of functional importance to the placenta [4,5,6,7,8,9,10,11,12,13]. Acquired 25–60 million years ago, respectively [14], both proteins have conserved specific expression in the placenta as well as features common to retroviral envelope proteins, i.e., a surface (SU) and a transmembrane (TM) subunit, a fusion peptide, an immunosuppressive domain, and a plasma membrane localization signal [15].

Several reports convincingly provided evidence that Syn-1 and -2 are necessary for the formation and renewal of the placental multinucleated syncytiotrophoblast (STB) layer [4,5,12,13,16]. This important structure is responsible for nutrient and gas exchanges between maternal and fetal circulations, as well as the release of hormones and cytokines, but needs to be continuously renewed through the fusion of underlying villous cytotrophoblast cells (vCTB) [17,18,19]. The fusion of vCTB involves the fusogenic activity of Syn-1 and -2, which depends on their interaction with specific receptors, identified as SLC1A4 and SLC1A5 for Syn-1 and MFSD2a for Syn-2 [5,20,21,22]. Most importantly, the decreased expression of Syn-1 and Syn-2 in the placenta was associated with pre-eclampsia (PE), a pregnancy syndrome characterized by a defect in placentation, endothelial cell dysfunction, and inflammation [11,23,24,25]. Another fusogenic envelope protein discovered in the placenta, namely, EnvP(b), was also investigated for a functional role, but was not demonstrated to be implicated in STB formation [26,27]. Similarly to Syn-1 and Syn-2, EnvP(b) also possesses immunosuppressive properties and could, nonetheless, participate in the tolerogenic immune response that prevails at the maternal–fetal interface during pregnancy, along with Syn-1 and Syn-2 [26,28,29,30,31].

Although hERV envelope proteins were mostly investigated in the context of the placenta, previous studies shed light on their potential implication in other physiological processes, such as myoblast and osteoclast fusion [32,33,34,35,36]. Furthermore, numerous reports have demonstrated that abnormal expression of these proteins was associated to various pathologies [37,38,39]. Indeed, several studies prominently associated the expression of Syn-1 and another ERV-W member with the physiopathology of multiple sclerosis (MS) [40,41,42,43,44,45]. Moreover, the abnormal expression of Syn-1 was reported in several cancers and was associated with the prognosis and cancer progression, by mediating the fusion of cancer cells with endothelial cells [46,47,48,49,50]. EnvP(b) was also shown to contribute to placental cancer progression, mediating fusion with endothelial cells [49]. Interestingly, Syn-1 and -2 are also expressed on the surface of placental extracellular vesicles [29,51,52]. The association of both proteins with these small vesicles might direct their cellular uptake through specific interaction with their receptors, possibly defining a cellular tropism for placental extracellular vesicles [53].

The β-galactoside-binding protein galectin-1 (Gal-1) is a small soluble protein displaying diverse functions in immunoregulation, angiogenesis, cell migration, embryo implantation, and placentation [54,55,56,57,58,59,60,61]. Gal-1 was the first identified member of the galectin protein family, which regroups 15 proteins that share a 130 amino acid region, called the carbohydrate-recognition domain (CRD) [62,63]. Gal-1 is synthesized as a monomer, containing a single globular CRD, and non-covalently associates into homodimers with two oppositely oriented CRDs that can functionally cross-link glycoproteins expressed on the same cellular membrane or on two adjacent cells [64]. In the placenta, Gal-1 expression localizes to extravillous cytotrophoblast cells (evCTB), vCTB, and STB and is involved in maternal–fetal immunotolerance, evCTB migration, and the fusion of vCTB [54,65,66,67,68]. Gal-1 is also an important factor in tumor progression and viral infections [55,61].

Based on the expression of Gal-1 in vCTB and STB and its previously reported impact on infection by Human Immunodeficiency Virus type-1 (HIV-1) and Human T-cell Leukemia Virus type-1 (HTLV-1) [69,70,71], we previously investigated the potential modulation of hERV Env-mediated trophoblast fusion [72]. Using Syn-2-pseudotyped viruses, we demonstrated that Gal-1 increased the infectivity of these viruses. Interestingly, we also found that Gal-1 decreased the infectivity of Syn-1-pseudotyped viruses. Based on this observation, the aim of the present study was to further investigate the association of Gal-1 with Syn-1, Syn-2, and EnvP(b) fusogenic properties in various cell lines. Our results mainly showed that Gal-1 increased the infectivity of Syn-2-pseudotyped viruses in all tested cell lines, whereas its association with Syn-1-pseudotyped viruses resulted in cell-specific outcomes. Interestingly, Gal-3, another placental galectin, had limited effect on both Syn-1- and Syn2-pseudotyped viruses.

## 2. Materials and Methods

### 2.1. Cell Lines

Human Embryonic Kidney (HEK) 293T, HeLa, Cos-7, Jurkat, and U87-MG were purchased from ATCC (Manassas, VA, USA), while HUVEC were kindly provided by Dr. Borhane Annabi (Université du Quebec à Montréal, Montreal, QC, Canada). HEK293T, HeLa, and Cos-7 were grown in Dulbecco’s Modified Eagle’s Medium supplemented with 10% fetal bovine serum (FBS) (Wisent Inc., St-Bruno, QC, Canada). U87-MG cells were maintained in Eagle’s Minimum Essential Medium (EMEM) supplemented with 10% FBS (Wisent Inc., St-Bruno, QC, Canada). HUVEC cells were grown in EGM-2 BulletKit medium (Lonza, Walkersville, MD, USA) containing 5% FBS. Jurkat cells were cultured in RPMI-1640 medium containing 10% FBS (Wisent Inc., St-Bruno, QC, Canada). All cells were maintained at 37 °C in a 5% CO_2_ atmosphere.

### 2.2. Plasmids

The pNL4.3env-luc vector was obtained from the NIH AIDS Reagent Program (Germantown, MD, USA). The vesicular stomatitis virus G protein expression vector, pLP/VSV-G, was purchased with the ViraPower Lentiviral Packaging Mix kit (K497500, Invitrogen, Thermo Fisher Scientific, Waltham, MA, USA). Codon-optimized cDNAs for Syn-1 and EnvP(b) were synthesized (GenScript, Hong Kong) and cloned between the HindIII and NotI sites of the pTH vector for Syn-1 or in frame with a V5-tag between the HindIII and BamHI sites of the pTH vector for EnvP(b). The pLVX-MSFD2aV5/His expression vector was generated following PCR amplification of the MSFD2aV5/His cDNA from the pEF6-MSFD2aV5/His expression vector [73] (a kind gift from Tommaso Dragani and Francesca Colombo, Instituto Nazionale Tumori, Milan, Italy) with the BamHI site containing forward 5′-CATCAGGAATTCGGATCCATGGCCAAAGGAGAAGGCGCCGAGAGC-3′ and PspOMI site containing reverse 5′-CCGAGCGGGCCCTCAATGGTGATGGTGATGATGACCG-3′ primers. The amplified product was then cloned in pLVX-Puro (Takara Bio Inc., Mountain view, CA, USA) using the same restriction sites. Single N-glycosylation MFSD2a mutants were generated by site-directed mutagenesis of the cDNA to convert asparagine 217 and 227 into glutamine codons using the Phusion High-Fidelity DNA Polymerase enzyme (New England Biolabs Ltd., Whitby, ON, Canada) and the following forward and reverse primers: N217Q 5′-GTTTCCAGGACCTCCAGAGCTCTACAGTAG-3′, N217Q 5′-AAGGCGTGTCTGCTTGGCCCACGATTTGTC-3′, N227Q 5′-CTTCACAAAGTGCCCAGCATACACATGGCAC-3′, and N227Q 5′-CTACTGTAGAGCTATTGAGGTCCTGGAAAC-3′. The phCMV1-eGFP, phCMV1-Syncytin-2, and phCMV1-Syncytin-2-Flag expression vectors were previously described [12,72].

### 2.3. Recombinant Galectin-1 and Galectin-3

Recombinant galectins were purified, as previously described [72]. Briefly, BL21(DE3)-hGal-1 or hGal-3 bacteria were inoculated in Terrific Broth and cultured overnight at 37 °C. Expression of the recombinant protein was induced by addition of 1 mM Isopropyl-β-D-thiogalactoside (IPTG). Bacterial pellets were resuspended in ice cold buffer (22 mM Tris-HCl pH 7.5, 5 mM EDTA, 1 mM DTT, and a protease inhibitor cocktail (Sigma-Aldrich, St. Louis, MO, USA) and lysed by sonication (30 s at 120 W (8 times; 1 min interval)). Lysates were cleared by ultracentrifugation at 112,500× *g* for 30 min at 4 °C (T70.1 rotor) in an L8-80 M centrifuge (Beckman Coulter Inc., Brea, CA, USA). Recombinant Gal-1 and Gal-3 were trapped on α-Lactose agarose column (Sigma-Aldrich, St. Louis, MO, USA), washed with PBS, and eluted in 1 mL fractions with 10 mL of 150 mM lactose (Sigma-Aldrich, St. Louis, MO, USA) in PBS. For Gal-1, fractions that contained the galectin were pooled and incubated overnight at 4 °C with 100 mM iodoacetamide to prevent oxidation of Gal-1. Free iodoacetamide and lactose were then removed by a series of dialysis against PBS. Fractions that contained Gal-3 were pooled, and lactose was removed using a HiPrep 26/10 desalting column (GE HealthCare, Chicago, IL, USA). Proteins were further passed on Acticlean Etox columns (Sterogene, Carlsbad, CA, USA) to remove endotoxins and then filter-sterilized using syringe filters (0.22 µm pore size) (Millipore, Cork, Ireland). Protein concentration was determined by the Bradford assay. Finally, endotoxin activity was assessed by the LAL assay (QCL-1000 Assay, Lonza, Mississauga, ON, Canada). Red blood cell hemagglutination assay was used to evaluate Gal-1 and -3 activities before use.

### 2.4. Production of Pseudotyped NL4.3 Viruses and Titration

The envelope-defective luciferase-expressing NL4.3 construct was used to produce CMVeGFP (env-control), Syn-2-, Syn-1, or VSV-G-pseudotyped virions. Using polyethylenimine (PEI) (Polysciences Inc., Warrington, PA, USA), HEK293T (2 × 10^6^) cells were co-transfected (1 µg DNA: 7 µL PEI (1 µg/µL PEI stock solution)) with 5 µg of pNL4.3env-Luc and 1.5 µg of either phCMV-Syncytin-2Flag, pTH-Syncytin-1-V5, phCMV-eGFP, or pLP/VSV-G. Cells were then replenished with fresh medium, which was next harvested between 36 and 40 h after transfection. Supernatants were centrifuged at 300× *g* for 5 min, passed through a 0.22 µm-syringe filter (VWR), aliquoted, and stored at −80 °C before use. Virus-containing supernatants (10 µL) were next quantified by the addition of 40 µL of a reaction mix (60 mM Tris-HCl pH 7.9, 6 mM MgCl_2_, 0.2 M KCl, 0.6 mM EGTA, 0.06% Triton X-100, 2.5% ethylene glycol, 6 mM DTT, 0.4 mM GSH, 0.025 U poly(rA) oligo-dT, and 2.5 µCi ^3^H-dTTP) and incubation for 2 h at 37 °C. The reaction was stopped with 150 µL of cold 10% trichloroacetic acid (TCA) at 4 °C for 30 min. Samples were then loaded onto a Millipore Multiscreen Glass fiber FC plate (Millipore-Sigma, Etobicoke, ON, Canada) and aspirated with a Millipore Multiscreen Manifold (Millipore-Sigma, Etobicoke, ON, Canada). Plates were washed once with 200 µL of 10% cold TCA and twice with 200 µL of cold 95% ethanol. Filters were transferred in scintillation vial containing 4 mL of scintillation solution. Reverse transcriptase activity was determined with the Tri-Carb 2800TR liquid scintillation analyzer and Quanta Smart software (PerkinElmer, Woodbridge, ON, Canada).

### 2.5. Infection with Pseudotyped Viruses

HEK293T, HeLa, Cos-7, and U87-MG (5 × 10^4^) and HUVEC (5 × 10^3^) cells were plated on 24-well plates 16 h before infection. Jurkat cells (10^6^) were seeded on 24-well plates in RPMI-1640 without FBS and incubated for 1 h at 37 °C before infection to allow adhesion of cells to the wells. Incubation of Jurkat cells in serum-free media was previously shown to allow transient adhesion of cells without altering their viability [74]. For experiments with wild-type MFSD2a or N-glycosylation N217Q and N227Q mutants, HeLa cells were transfected with 0.5 µg of pLVX-MFSD2aV5, pLVX-N217QV5, or pLVX-N227QV5 expression vectors and 3 µL of the lipofectamine 2000 reagent (Invitrogen, Carlsbad, CA). Cells were infected with fixed quantity of virions (adjusted according to measured reverse transcriptase activities of virus stocks) in a final volume of 200 µL and spinoculated at 1000× *g* for 2 h at 25 °C. After infection, cells were rinsed once in PBS and replenished with complete media (Wisent Inc., St-Bruno, QC, Canada). Cells were analyzed 24 h post-infection through luciferase assays, as previously described [72]. Luciferase activities were calculated as the mean Relative Light Unit (RLU) value ± standard error of the mean (SEM) of triplicate samples and normalized against the protein concentration of each sample.

### 2.6. Computational Analysis of MFSD2a Transcript Expression in Human Cell Lines

MFSD2a expression levels in HeLa, HEK293T, HUVEC, and U87-MG were obtained from the Human Protein Atlas database in the Cell Atlas (www.proteinatlas.org/ENSG00000168389-MFSD2A/cell, accessed on 1 October 2023). The RNA-sequencing data for MFSD2a, expressed as transcripts per kilobase million (TPM), were plotted against mean relative light unit (RLU) values obtained from three independent infection experiments of HeLa, HEK293T, HUVEC, and U87-MG cells with Syn-2-pseudotyped viruses.

### 2.7. MTT Assay

Cells were seeded on 24-well plates before incubation with either PBS, 4 µM recombinant Gal-1, or 50 mM lactose (Sigma-Aldrich, St. Louis, MO, USA) for 2 h and centrifuged at 1000× *g* to optimize infection conditions. Cells were then washed once with PBS and incubated at 37 °C in 5% CO_2_ for 24 h after addition of complete media. Medium was removed, and thiazolyl blue tetrazolium bromide (2.5 mg/mL final concentration) (M5655 Sigma-Aldrich, St. Louis, MO, USA) was subsequently added to cells incubated at 37 °C for 3 h. Following incubation, 300 µL of MTT solvent solution (4 mM HCl and 0.1% NP40 in isopropanol) was added, and plates were incubated for 30 min at room temperature (RT). Supernatants (100 µL) were transferred onto a 96-well plate in triplicate, and the absorbance (OD_570 nm_) was measured with a microplate spectrophotometer (Eon Spectrophotometer, BioTek instruments, Winooski, VT, USA).

### 2.8. Cell Fusion Assay

HEK293T cells (5 × 10^4^) were seeded on glass coverslips 16 h prior to transfection with control, EnvP(b)-, or Syn-2-expression vectors using PEI as a transfection reagent (1 µg DNA:7 µL PEI ratio). For experiments with Syn-2, cells were transfected with 0.35 µg of plasmids (phCMV1- or phCMV1-Syncytin-2). For experiments with EnvP(b), 0.5 µg of the pTH- or pTH-EnvP(b) plasmids were used. Six hours after transfection, cells were incubated with either PBS, 4 µM recombinant Gal-1, or 4 µM Gal-1 and 50 mM lactose for 24 h at 37 °C. Plasma membranes and nuclei were then stained using CellMask Orange (1 µg/mL) and Hoechst (2 µg/mL) (Thermo Fisher Scientific, Waltham, MA, USA), respectively, following manufacturer’s instructions. Cells were fixed for 15 min in 2% PFA in PBS at RT and mounted on a drop of ProLong Gold antifade reagent (Invitrogen, Thermo Fisher Scientific). Cells were then visualized by laser-scanning confocal microscopy with a Nikon Eclipse Ti microscope coupled with a Nikon A1R confocal unit and Plan Fluor 20×/0.75 Mlmm oil or CFI Plan Apochromat λ 60×/1.4 N.A. objectives (Nikon Canada, Mississauga, ON, Canada). A syncytium was defined as a cluster of at least three nuclei in the same cytoplasm [25]. For each condition, eight microscopic fields were analyzed, and a cellular fusion index was calculated by dividing the number of nuclei per syncytia by the total number of nuclei in one field and multiplying the result by 100. Over 1500 nuclei were counted per condition. An average percentage was then calculated, and the experiment was repeated three times for both Syn-2- or EnvP(b)-transfected cells. A final cellular fusion index (mean +/− SEM) was calculated and represents the final average of three values obtained from three independent experiments.

### 2.9. Immunolocalization of MFSD2a

Cos-7 cells (4 × 10^4^) were plated on glass coverslips on 24-well plates. Cells were transfected with 0.5 µg of empty vector or expression vectors for wild-type MFSD2a or N-glycosylation mutants using PEI (1 µg DNA: 7 µL PEI ratio). Forty-eight hours after transfection, cells were fixed with 2% PFA in PBS for 15 min at RT and then washed three times in PBS. Cells were permeabilized for 15 min in PBS 0.01%/Triton X-100 at RT and washed three times in PBS. Blocking solution (5% FBS and 5% BSA in PBS) was added for 45 min at RT, and, after addition of mouse monoclonal anti-V5 antibodies (1/500, R960-25, Thermo Fisher Scientific), cells were incubated overnight at 4 °C with shaking. Following three washes in PBS, cells were incubated with Alexa Fluor 568-conjugated goat anti-mouse IgG antibodies (1/1000, A11004, Invitrogen, Thermo Fisher Scientific) for 1½ h at RT. Cells were then washed three times in PBS, treated with DAPI NucBlue Cell Stain ReadyProbes reagent (Thermo Fisher Scientific) for 45 min at RT, and finally mounted on a drop of ProLong Gold antifade reagent (Invitrogen, Thermo Fisher Scientific) after three washes in PBS. Signals were analyzed by confocal microscopy using the Eclipse Ti microscope coupled with an A1R confocal unit and the CFI Plan Apochromat λ 60×/1.4 objective (Nikon Canada, Mississauga, ON, Canada).

### 2.10. Western Blot Analysis

Cellular extracts (25 µg) were resolved on 10% acrylamide/bis-acrylamide SDS gels. Proteins were transferred on 0.45 µm methanol-activated Amersham Hybond PVDF membranes (GE Healthcare Life Science, Freiburg, Germany). Membranes were incubated with a blocking solution (5% skimmed-milk and 0.05% PBS-Tween20 (PBST)) for 1 h at RT. Mouse monoclonal anti-V5 antibodies (1/3000, R960-25, Thermo Fisher Scientific) or mouse monoclonal anti-GAPDH antibodies (1/2000, sc32233, Santa-Cruz Biotechnology Inc., Santa-Cruz, TX, USA) were then incubated overnight at 4 °C. Following three washes in PBST, membranes were incubated with ECL horseradish peroxidase-conjugated sheep anti-mouse IgG secondary antibodies (1/5000, GE Healthcare). After three washes in PBST, proteins were detected in 10 mL of revealing solution (Sol. A: 0.1 M Tris-HCl pH 10.5 and 18% H_2_O_2_ mixed in Sol. B: 0.1 M Tris-HCl pH 10.5, 450 µM p-Coumaric acid diluted in DMSO, and 7.5 mM Luminol in DMSO) using the Fusion Fx7 apparatus (Montreal Biotech Inc., Dorval, QC, Canada).

### 2.11. Statistical Analysis

Otherwise indicated in figure legends, results were obtained from three independent experiments. Graphpad Prism 6 was used for all statistical analyses. Comparison of multiple means was achieved with 1-way or 2-way ANOVA tests, followed by Bonferroni’s multiple comparison tests. 

## 3. Results

### 3.1. Galectin-1 Increases the Fusion Capacity of the Endogenous Retroviral Envelope Proteins Syncytin-2 and EnvP(b)

We previously established a link between Syn-2 and Gal-1 using Syn-2-pseudotyped viruses and showed that the addition of recombinant Gal-1 during the infection of HEK293T cells significantly increases the infectivity of pseudotyped viruses [72]. The combination of lactose with Gal-1, an antagonist of several galectin proteins [75], precluded the effect of Gal-1, showing a CRD-dependent effect of Gal-1 during infection. 

We first wished to confirm this Gal-1-dependent effect on Syn-2-mediated fusion in a cell fusion assay using HEK293T cells, which form syncytia when overexpressing Syn-2 [6,26]. In parallel, we were also interested to test if Gal-1 could act on cell fusion mediated by the hERV-encoded EnvP(b), as we were not able to produce infectious pseudotyped viruses with this envelope [26,27,31]. After transfection, HEK293T cells were incubated with PBS or Gal-1 in the presence or absence of lactose, and the fusion of Syn-2- or EnvP(b)-expressing cells was assessed by confocal microscopy (Figure 1A,B). In these experiments, cells were transfected with a low amount of Syn-2 in order to better appreciate the potential effect of Gal-1 during the fusion. The results presented in Figure 1 showed that for both Syn-2 and EnvP(b), Gal-1 treatment increased the number of syncytia compared to the controls. Also, the addition of lactose reduced the number of syncytia to the level of the PBS control, showing a CRD-dependent effect. Measurement of the cell fusion index confirmed these results: in the presence of Gal-1, the fusion index significantly increased to 17.5% +/− 2.1 (vs. 8.56% +/− 0.6 for control) (*p* = 0.016) for Syn-2 (Figure 1C) and to 18.2% +/− 7.5 (vs. 4.29% +/− 1.3 for the PBS control) for EnvP(b) (Figure 1D). When 50 mM lactose was added with Gal-1, the fusion index reached 5.76% +/− 0.22 for Syn-2 (Figure 1C) and 3.02% +/− 1.4 for EnvP(b) (Figure 1D). Thus, extracellular Gal-1 potentiates the fusion capacity not only of Syn-2 but also of EnvP(b), an effect that depends on the glycan-binding capacity of this galectin.

### 3.2. Syncytin-1, Syncytin-2, and VSV-G-Pseudotyped Viruses Differentially Infect Human and Simian Cell Lines 

We next wanted to analyze the impact of Gal-1 on Syn-1 and -2 in various cell types, some of which are known to have a potential functional association with Syncytin proteins. Cells lines included HEK293T and Cos-7, which both express human and simian MFSD2a, respectively, and non-MFSD2a-expressing HeLa cells [6,12,21]. We also tested HUVEC and Jurkat cell lines as models of endothelial and CD4+ T lymphocytes, having been associated with Syncytin-mediated fusion and immunosuppression, respectively [29,30]. We also added U87-MG glioblastoma cells, as Syn-1 has been associated with various neurodegenerative diseases in which various CNS-residing cells (including glial cells) have been potentially implicated [39]. 

We first tested the infectivity of our pseudotyped viruses on these cells. As expected, env-control viruses were non-infectious on all tested cell lines, presenting values similar to mock-infected cells (Figure 2A). High levels of variations in infection rates were observed between the three pseudotyped viruses, with VSV-G-pseudotyped viruses being more infectious than Syncytin-pseudotyped viruses, as expected (Figure 2B–D). Using the Human Protein Atlas Cell Atlas database (www.proteinatlas.org/ENSG00000168389-MFSD2A/cell), we collected data related to MFSD2a mRNA expression levels on available cell lines and compared these data with the infection values obtained with Syn-2-pseudotyped viruses (Figure 2E). We, indeed, observed a direct relation between the infectivity of Syn-2-pseudotyped viruses and MFSD2a transcripts levels for the HeLa, HUVEC, HEK293T, and U87-MG cell lines, indicating that our model of pseudotyped viruses is representative of the interaction between Syn-2 and MFSD2a. A correlation for Syn-1-pseudotyped viruses was more difficult to assess, likely due to the presence of varying expression levels of the two receptors, SLC1A4 and SLC1A5.

### 3.3. Increase in the Infectivity of Syncytin-2-Pseudotyped Viruses by Galectin-1 in Different Cell Lines Is MFSD2a-Dependent

We next wanted to confirm the Gal-1-dependent modulation of the infectivity of Syn-2-pseudotyped viruses in these cell lines. We first looked at the effect of Gal-1 and lactose treatment on cell viability and found that neither condition significantly altered the survival of HEK293T, HeLa, or Cos-7 cells (Figure 3A). We then compared the infectivity rates of Syn-2-pseudotyped viruses on HEK293T and Cos-7 cell lines in the presence of 4 µM Gal-1 (Figure 3B). A significant increase in infectivity was observed when Gal-1 was added, compared to the virus-only condition, on both HEK293T (5.51-fold +/− 0.76, *p* = 0.001) and Cos-7 cells (6.93-fold +/− 1.54, *p* = 0.0074). Lactose abolished the effect of Gal-1, showing a dependency on the availability of the CRD.

As Gal-1 was found to bind N-glycans on HIV-1 gp120 and the CD4 receptor [76], we tested if the N-glycosylation status of MFSD2a was important for the effect of Gal-1 during infection. MFSD2a possesses two N-glycosylation sites, asparagine 217 and 227 [77,78], which were both individually mutated to generate two MFSD2a asparagine-to-glutamine mutants (N217Q and N227Q). When a double mutant was generated, weaker expression and plasma membrane localization unfortunately prevented us from including this mutant in our experiments. After confirming the expression and plasma membrane localization of two other mutants (Appendix A), expression vectors for wild-type MFSD2a and the mutated versions, N217Q and N227Q, were transfected in HeLa cells, which do not express endogenous MFSD2a. HeLa cells were then infected with Syn-2-pseudotyped viruses in the presence or absence of Gal-1. Our results confirmed the necessity for MFSD2a expression, as Gal-1 had no effect on mock-transfected cells but induced a significant 9.3-fold increase in infectivity on MFSD2a-expressing cells (9.3 +/− 1.52) (Figure 3C). Interestingly, when N217Q- or N227Q-expressing HeLa cells were infected, a 10-fold increase in infectivity was observed compared to control conditions (N217Q, 10.8-fold +/− 3.96; N227Q, 10.2-fold +/− 4.75). In each experiment, the addition of lactose counteracted the effect of Gal-1 on the infectivity of the viruses. Thus, the depletion of either N-glycan did not alter the impact of Gal-1 on the infectivity of Syn-2-pseudotyped viruses.

Altogether, these results show that the effect of Gal-1 on the infectivity of Syn-2-pseudotyped-viruses is not cell-type-specific (and can act on simian cell line) and does not depend on the specific glycosylation status of one asparagine residue.

### 3.4. Galectin-1-Dependent Increase in the Infectivity of Syncytin-2-Pseudotyped Viruses Is Reproduced on Human Cell Lines of Different Types

We next investigated the association between Gal-1 and Syn-2 in other human cell lines (shown in Figure 2). As Gal-1 was previously shown to induce apoptosis in Jurkat and HUVEC cells [79,80], the viability of HUVEC, Jurkat, and U87-MG cells was, thus, first evaluated for the tested concentrations (Figure 4A). No significant impact on the cell viability was noted upon Gal-1 treatment for any of the tested cell lines. We then studied the effect of Gal-1 on infection and found a significant increase in the infectivity of Syn-2-pseudotyped viruses (Figure 4B–D) (Jurkat, 14.1 +/− 1.75; HUVEC, 33.4 +/− 0.05; U87-MG, 6.76 +/− 1.29). The addition of lactose with Gal-1 inhibited this increase in infectivity. 

These results showed that the effect of Gal-1 on the infectivity of Syn-2-pseudotyped viruses is reproduced in several human cell lines and suggested that the presence of Gal-1 in the extracellular space potentiates the interaction between Syn-2 and MFSD2a.

### 3.5. Response of the Infectivity of Syncytin-1-Pseudotyped Viruses to Galectin-1 Is Dose-Dependent and Varies in Different Cell Lines

Our recent results showed that Gal-1 reduced the infection of HEK293T cells by Syn-1-pseudotyped viruses [72]. We were interested in further testing the effect of Gal-1 on the infection of our selected cell lines by these viruses (Figure 5). When the infectivity of Syn-1-pseudotyped viruses was compared, we first confirmed a significant decrease in the infectivity on HEK293T in the presence of 4 µM of Gal-1 (Figure 5A; mean fold change 0.42 +/− 0.04). Interestingly, a significant increase in the infectivity was measured in HeLa and Cos-7 cells for the same dose (Figure 5B,C, 14.6-fold +/− 2.49 and 8.77-fold +/− 1.45, respectively). When Jurkat and HUVEC cells were infected in the presence of Gal-1 (Figure 5D,E), the infectivity was also significantly increased when compared to control (8.1-fold +/− 0.19 and 24.8 +/− 5.5, respectively). In contrast, the addition of Gal-1 to U87-MG did not significantly change the infectivity compared to the virus-only condition, despite a slight increment (Figure 5F, 1.59-fold +/− 0.36). All the observed Gal-1-dependent effect in the above experiments was specifically blocked by the addition of lactose.

When comparing the infection rates of HEK293T and U87-MG vs. HeLa, HUVEC, and Cos-7 cells (Figure 2B), we noticed that the first two cell lines had higher RLU counts than the three other cell lines. We hypothesized that the 4 µM dose might not be optimal for Syn-1-pseudotyped viruses on all cell lines, depending on the susceptibility of the cells to infection. We, thus, tested lower doses of Gal-1 during the infection of HEK293T and U87-MG and compared them to HeLa cells (Figure 6). Interestingly, we observed a significant increase in the infectivity with doses ranging from 0.1 µM to 1 µM for HEK293T cells (Figure 6A) and 1.5 µM (U87-MG; Figure 6B) followed by a decrease in the infectivity with higher doses. On HeLa cells, the infectivity of Syn-1-pseudotyped viruses progressively increased to reach a plateau at 1.5 µM (Figure 6C). In each experiment, the addition of lactose inhibited the effect of Gal-1 (tested with 2 µM Gal-1: HEK293T (51.1 +/− 0.15 RLU); U87-MG (55.4 +/− 0.87 RLU); HeLa 48.0 +/− 8.55 RLU)).

Altogether, these results demonstrate that infection of cell lines by Syn-1-pseudotyped viruses is differently affected by Gal-1 and that cells more prone to infection optimally respond at a lower concentration.

### 3.6. Galectin-3 Does Not Increase the Infectivity of Syncytin-2-Pseudotyped Viruses but Affects Syncytin-1-Pseudotyped Viruses in Certain Cell Lines

In addition to Gal-1, other members of the galectin family are expressed in the placenta [81]. Among placental galectins, Gal-3 was extensively studied, localizes to villous and extravillous cytotrophoblast cells, and is expressed in the choriocarcinoma BeWo cell line [65,82,83]. We wished to address its impact on the infectivity of Syn-1- and Syn-2-pseudotyped viruses toward different cell lines. HEK293T, HeLa, and U87-MG cells were, thus infected, with pseudotyped viruses in the absence or presence of Gal-3 (Figure 7). HeLa infection experiments with Syn-2-pseudotyped viruses were conducted either in mock-transfected cells or cells transfected with a MFSD2a expression vector. Our results showed that Gal-3 had no effect on the infectivity of Syn-2-pseudotyped viruses in any of the tested cells (Figure 7A–C). When Syn-1-pseudotyped viruses were tested, Gal-3 significantly increased the infectivity of HeLa cells in a lactose-sensitive manner. However, Gal-3 did not significantly alter the infection of HEK293T or U87-MG cells, despite demonstrating a non-significant increase in the latter cell line (Figure 7D–F). 

These results demonstrated that Gal-3 does not modulate infection of the tested cell lines by Syn-2-pseudotyped viruses but increases the infectivity of Syn-1-pseudotyped viruses in certain cell lines.

## 4. Discussion

In a recent study, we described a specific association between Syn-2 and Gal-1 using Syn-2-pseudotyped viruses and showed an interaction between Gal-1 and Syn-2-pseudotyped viral particles, resulting in increased infectivity. Our results also suggested that the interaction of Gal-1 with Syn-2-pseudotyped viruses maximized the interaction between Syn-2 and MFSD2a, as the effect was only observable during the early phase of infection [72]. In the present study, we extend this association in different cell lines and provide evidence that, using cell fusion and infection assays, Gal-1 increased the binding of Syn-2 with MFSD2a, likely by binding two both cell surface proteins and creating a bridge. In each experiment, the presence of lactose, an antagonist to Gal-1 that competes for the binding of the CRD, inhibited the activity of Gal-1. This suggests that the associations among Gal-1, Syn-2, and MFSD2a entail a protein–glycan interaction, which was reported to involve N-glycans in several other studies showing an interaction between Gal-1 and viral envelopes [76,84,85,86].

The presence of two and nine N-glycosylation sites was previously confirmed for MFSD2a and Syn-2, respectively [78,87]. Nevertheless, our results indicate that Gal-1 is likely not specific to a single MFSD2a N-glycan. Indeed, the expression of two MFSD2a versions mutated at a single different N-glycosylation site in HeLa cells did not change the effect of Gal-1 on the infectivity rate of Syn-2-pseudotyped viruses. Unfortunately, the double N-glycosylation mutant, in which both asparagines were mutated, was poorly expressed and mostly retained in the endoplasmic reticulum and could not be used in our infection experiments. Thus, it is possible that either of the N-linked glycosylated asparagine residue is sufficient for the binding of Gal-1 to MFSD2a. Other possible explanations for these results could rely on the binding of Gal-1 to O-glycans or the Gal-1-mediated clustering/stabilization of Syn-2 homotrimers without direct interaction with the receptor, thus optimizing Syn-2/MFSD2a association or the TM-mediated fusion of adjacent membranes [88]. 

Our previous work showed that Syn-1 and -2 were differently affected by Gal-1, with a significant reduction in the infectivity of Syn-1-pseudotyped viruses on HEK293T cells compared to a significant increase for Syn-2-pseudotyped viruses [72]. The present results now demonstrate a dose-dependent effect of Gal-1 on the infectivity of Syn-1-pseudotyped viruses. At this point, cell-type specificity, although a possibility, cannot be confirmed, as only one cell type was tested per cell type. The observed variation on the impact of Gal-1 in different tested cell lines cannot be accounted by viral production, as the same preparation of Syn-1-pseudotyped viruses was used in different infection experiments. However, as Syn-1 can use two cellular receptors, SLC1A4 and SLC1A5 [20], the differential expression of these surface proteins between cells or differences in the composition or structure of N- or O-linked oligosaccharides from one cell line to another could explain these variations [89,90,91]. The latter hypothesis suggests the different post-translational glycosylation of SLC1A4 and SLC1A5, which possess two conserved N-glycosylation sites [20]. In addition, to our knowledge, the binding affinity for either one of these proteins was never investigated. Thus, it is possible that the interaction of Syn-1 with one receptor is more optimal, and variations in the expression of this receptor between cells could explain discrepancies in the effect of Gal-1. In contrast, the state of glycosylation on the different receptors might differ between cell lines, which could further affect the capacity of Gal-1 to efficiently bind [92]. Another mechanism that might account for differences in the sensitivity of cell lines toward infection by Syn-1-pseudotyped viruses in the presence of Gal-1 might rely on the variation in the abundance of other cell surface proteins of glycolipids inducing cell signaling following binding to Gal-1, although this is less likely, as it should have similarly impacted Syn-2-pseudotyped virus infection.

The dose-dependent response of Gal-1 to the infectivity of Syn-1-pseudotyped viruses toward HEK293T and U87MG, however, brings a different perspective to Gal-1 interaction with this HERV envelope. As infection of these cell lines is higher than other cell lines, it is interesting to note that only low concentrations of Gal-1 lead to an increase in the infectivity, while higher concentrations rather cause an important reduction. This is in contrast with results obtained with HeLa cells showing continuous induction at various concentrations, reaching a final plateau. We speculate that cells, which are a better target for infection by Syn-1-pseudotyped viruses, can only increase their susceptibility at a low concentration, allowing more beneficial and optimal binding to one of the receptors, while a higher concentration might divert Syn-1 binding to the less potent fusion-inducing receptor. Further experiments will be required to better define the receptors (SLC1A4, SLC1A5, or others) implicated in this proposed model.

Our results also suggest that Syn-2 specifically associates with Gal-1, as Gal-3 did not increase the infectivity of Syn-2-pseudotyped viruses on any tested cell lines, even those that express high levels of MFSD2a. These results are similar to previous reports on HIV-1 and confirm results from our previous study on Syn-2-pseudotyped viruses [71,72,76]. However, when Gal-3 was tested along with Syn-1-pseudotyped viruses, variation in the effect on the different cell lines was specifically noted. We observed a similar increase in the infectivity between Gal-1 and Gal-3 on HeLa cells, but the addition of the latter during infection did not change the infectivity on HEK293T cells. On U87-MG, Gal-1 increased the infectivity of Syn-1-pseudotyped viruses, although non-significantly, whereas Gal-3 did not modulate the infectivity. The cell-line-specific effect of Gal-1 or Gal-3 on the same virus has never been reported, to our knowledge, and we cannot rule out that the observed variations are inherent to the nature of the transformed cells used in these experiments. Again, Syn-1-pseudotyped viruses might be differently affected by their infection capacity in different tested cell lines, which would be based on the receptors with which they interact and how Gal-1 vs. Gal-3 helps bridge Syn-1 with these different receptors. Analyses are underway to molecularly characterize the interaction between Gal-1 (vs. Gal-3) and Syn-1/Syn-2, such as determining the nature and composition glycans of Syncytin proteins and their receptors and identifying which oligosaccharides are necessary for the binding of Gal-1.

Finally, using a cellular fusion assay, we found that Gal-1 potentiates the fusion activity of EnvP(b). The ectopic expression of EnvP(b) in HEK293T cells led to a significant increase in the syncytia number, when recombinant Gal-1 was added into culture media compared to control conditions, an effect that depended on the β-galactoside-binding activity of Gal-1. Previous reports identified EnvP(b) as a fusogenic ERV, transcriptionally expressed in several tissues, including the placenta, and that is involved in the fusion of trophoblastic cells, although it is not essential for syncytialization [26,27,49]. Interestingly, Aagaard et al. (2012) found that EnvP(b) mediated heterotypic cell fusion between BeWo and endothelial cells and, thus, could participate in tumor development [49]. Finding an association between Gal-1 and EnvP(b) is of particular interest, as many tumor cells secrete higher levels of Gal-1, which was found to stimulate tumoral angiogenesis and favor tumoral immune evasion [93,94,95]. Because EnvP(b) transcripts have been detected in many tissues, e.g., breast, ovaries, testis, skin, and colon, the higher secretion of Gal-1 by tumor cells could also favor EnvP(b)-mediated heterotypic tumor cell fusion and contribute to tumor heterogeneity [96,97]. As EnvP(b) possesses a functional ISD, we can also speculate that it could be involved in tumoral immune evasion. However, EnvP(b) protein expression should be confirmed in other tissues than the placenta, and expression levels should be compared with neoplastic tissues as a starting point. 

As Syn-1 and Syn-2 are expressed on circulating placental extracellular vesicles [29,51,98], the observed associations between galectins and Syn-1 and -2 could have implications during the interaction of these vesicles with cells, especially regarding endothelial and immune cells [99,100,101]. Moreover, the abnormal expression of Syn-1 was reported in several tumor cells and linked to the fusion between cancer and endothelial cells [46,47,49,50], a process that could also involve Gal-1, as the expression and secretion of this galectin is induced in the endothelial cells that surround tumors to stimulate tumoral angiogenesis [94,102].

## 5. Conclusions

In this study, we report the modulation of the fusogenic property of three placental endogenous retroviral envelope proteins, namely, Syn-1, Syn-2, and EnvP(b), by the β-galactoside-binding protein Gal-1. More specifically, we confirm the specific impact of Gal-1 on Syn-2, which potentiates the fusogenic capacity of Syn-2 in a CRD-dependent manner in several cell lines. We also observe a functional association between Syn-1 and two placental galectins, Gal-1 and Gal-3, with the outcome of these interactions being cell-type- and dose-dependent. Finally, our results suggest that EnvP(b) could also benefit from the presence of extracellular Gal-1. The fact that Gal-3 presents an effect with Syn-1- but not Syn-2 suggests that the conformation and/or the glycan composition of both proteins (or their receptors) is different and that their fusogenic activities could be differently regulated by extracellular Gal-1 and Gal-3. Confirming an association between placental galectins and Syn-1 and -2 brings new insights on the regulation of cell fusion. These results are also relevant regarding the biological function of placental extracellular vesicles. Our results suggest that the effect of Gal-1 could be relevant in several physiological processes involving HERV Env proteins, such as cytotrophoblast, myoblast, and osteoclast fusion; immune tolerance during pregnancy; and pathological conditions, such as multiple sclerosis and cancer, in which these fusogenic proteins show abnormal expression. 

This study brings new insights on the molecular events occurring during the syncytialization of trophoblast cells and suggests different regulatory mechanisms of the fusogenic activity of Syncytin proteins.

## Figures and Tables

**Figure 1 viruses-15-02441-f001:**
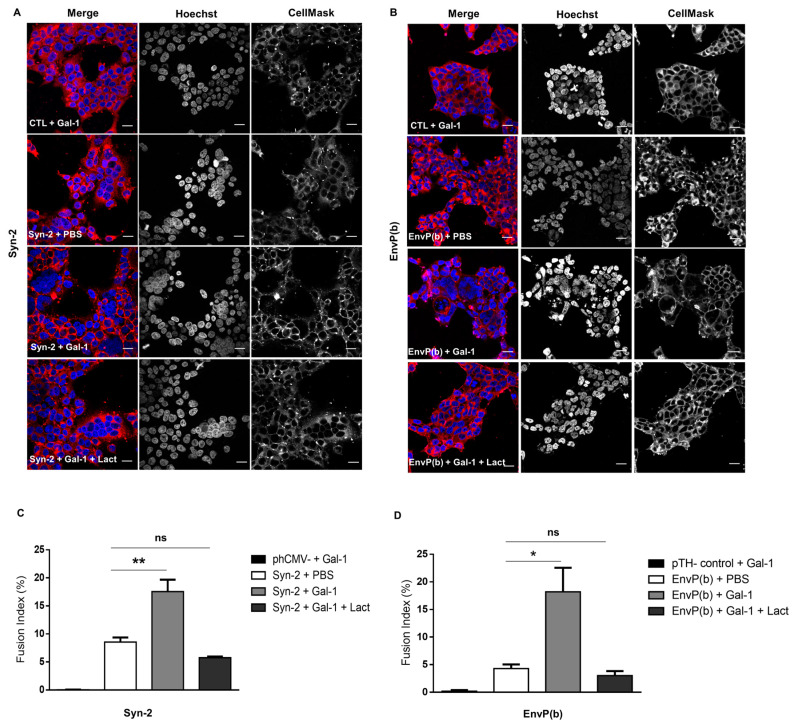
Galectin-1 increases the fusion of Syncytin-2- and EnvP(b)-transfected HEK293T cells. (**A**,**B**) Confocal microscopy imaging of HEK293T cells transfected with phCMV (**A**) or pTH (**B**) empty vectors (CTL) or phCMV-Syn-2 (**A**) or pTH-EnvP(b) (**B**) expression vectors and treated with PBS, 4 µM Gal-1, or 4 µM Gal-1 + 50 mM lactose for 24 h. Plasma membranes and nuclei were stained with CellMask and Hoechst solutions, respectively, and then cells were fixed in 2% PFA solution. Cell fusion was assessed by confocal microscopy imaging with an oil 60× objective. Scale bar: 20 µm. (**C**,**D**) Cellular fusion index calculated from Syn-2- and EnvP(b)-transfected HEK293T cells. For each condition, syncytial nuclei (Ns) and total number of nuclei (Nt) from 8 confocal microscopy fields were counted, and a cell fusion index was calculated as follow: (Ns/Nt) × 100. The graph represents the mean fusion index +/− standard error of the mean (SEM) of three independent experiment, expressed as a percentage. * *p* < 0.05, ** *p* < 0.01; ns: non-significant.

**Figure 2 viruses-15-02441-f002:**
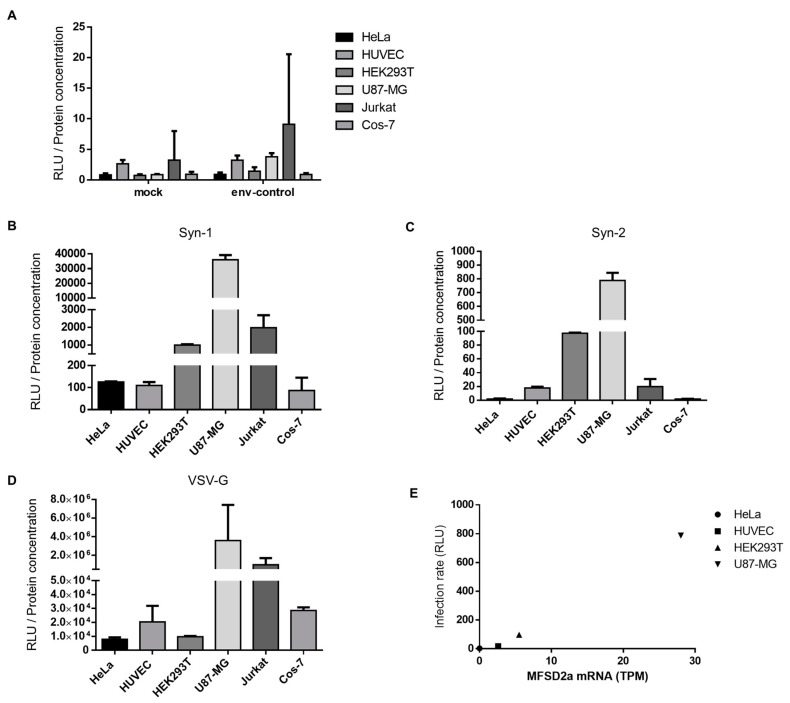
Infection rates of pseudotyped viruses on different cell lines. (**A**–**D**) Env-control (**A**), Syn-1-pseudotyped (**B**), Syn-2-pseudotyped (**C**), and VSV-G-pseudotyped (**D**) viruses were used to infect five human and one simian cell lines in triplicate. Cells were lysed 24 h post infection (p.i), and luciferase activity was measured. Results are expressed as the mean relative light unit (RLU) +/− standard deviation (SD) normalized against protein concentration of samples of three independent experiments. (**E**) Luciferase activities associated to infection by Syn-2-pseudotyped viruses were plotted against MFSD2a transcript levels of HeLa, HUVEC, HEK293T, and U87-MG cells. MFSD2a expression levels are expressed as transcripts per kilobase million (TPM) and were obtained from the Human Protein Atlas Cell Atlas database. Infection rates are derived from three independent infection experiments.

**Figure 3 viruses-15-02441-f003:**
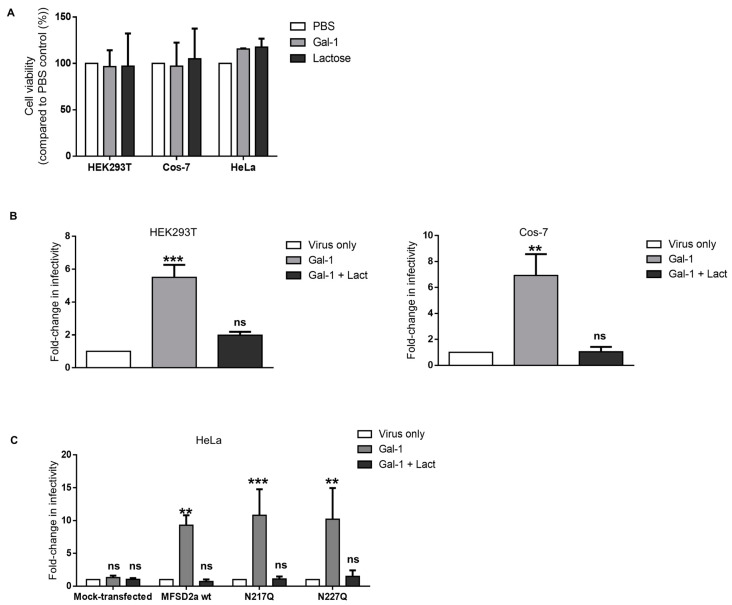
Gal-1 increases the infectivity of Syn-2-pseudotyped viruses on human and simian cells. (**A**) The MTT assay was performed on HEK293T, Cos-7, and HeLa cells incubated with PBS, 4 µM Gal-1, or 50 mM lactose. Results from two independent experiments were pooled and expressed as the mean percentage of viable cells compared to the control +/− SD. (**B**,**C**) HEK293T, Cos-7 (**B**), and HeLa (**C**) cells were infected by Syn-2-pseudotyped viruses in the presence of PBS (virus only), 4 µM Gal-1 (Gal-1), or 4 µM Gal-1 + 50 mM lactose (Gal-1 + Lact). (**D**) HeLa cells were transfected with either an empty vector (mock) or with expression vectors for wild-type MFSD2a, N217QMFSD2a, or N227QMFSD2a mutants. Luciferase activities were measured 24 h p.i. and normalized against protein concentration. Results are the mean +/− SEM of three independent infections, expressed as fold-change in infectivity compared to the virus-only condition. *** *p* < 0.001, ** *p*< 0.01; ns: non-significant (see Appendix A for raw data).

**Figure 4 viruses-15-02441-f004:**
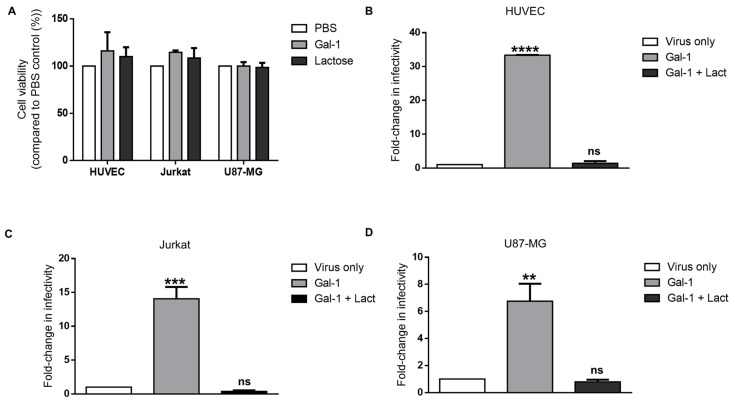
Gal-1 significantly increases the infectivity of Syn-2-pseudotyped viruses on different human cell lines. (**A**) The MTT assay was performed on HUVEC, Jurkat, and U87-MG cells incubated with PBS, 4 µM Gal-1, or 50 mM lactose. Results of two independent experiments are presented and are expressed as the mean percentage +/− SD of viable cells compared to control. (**B**–**D**) HUVEC, (**B**), Jurkat (**C**), and U87-MG (**D**) cells were infected by Syn-2-pseudotyped viruses in the presence of PBS (virus only), 4 µM Gal-1, or 4 µM Gal-1 + 50 mM lactose. Following infection, cells were lysed, and luciferase activities were measured and normalized against protein concentration. Graphs represent the mean fold-change in infectivity +/− SEM over the virus-only control of two or three independent infection experiments. ** *p* < 0.01, *** *p* < 0.001, **** *p* < 0.0001; ns: non-significant (see Appendix A for raw data).

**Figure 5 viruses-15-02441-f005:**
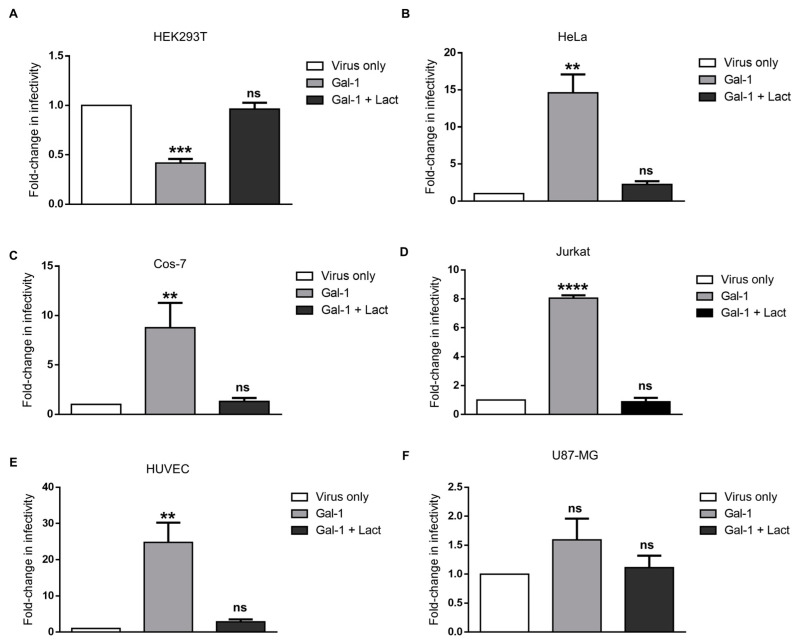
The modulation of the infectivity of Syn-1-pseudotyped viruses by Gal-1 is cell-line-dependent. HEK293T (**A**), HeLa (**B**), Cos-7 (**C**), Jurkat (**D**), HUVEC (**E**), and U87-MG (**F**) cells were infected with Syn-1-pseudotyped viruses in presence of PBS (virus only), 4 µM Gal-1 (Gal-1), or 4 µM Gal-1 and 50 mM lactose (Gal-1 + Lact). Cells were lysed 24 h p.i., and luciferase activities were measured in triplicate. For each sample, luciferase values were normalized against protein concentration, and results are expressed as fold-change in infectivity compared to PBS control (virus only). Graphs represent the average fold-change +/− SEM of three independent infection experiments for each cell line. ** *p* < 0.01, *** *p* < 0.001, **** *p* < 0.0001; ns: non-significant (see Appendix A for raw data).

**Figure 6 viruses-15-02441-f006:**
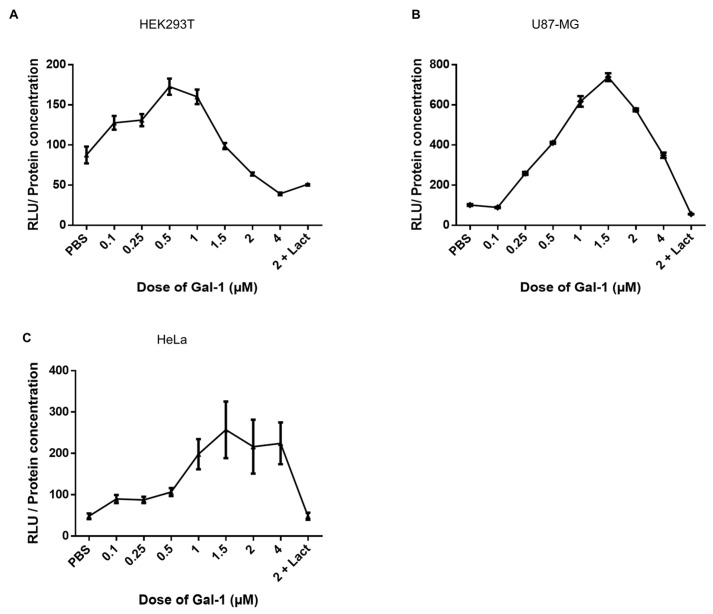
Dose response of galectin-1 on the infectivity of Syncytin-1-pseudotyped viruses depends on the cell line. HEK293T (**A**), U87-MG (**B**), and HeLa (**C**) cells were infected with Syn-1-pseudotyped viruses in the presence of PBS or increasing concentration of Gal-1 (0.1–4 µM). Cells were lysed 24 h p.i., and luciferase activity was measured in triplicate. For each sample, luciferase values were normalized over the protein concentration, and the results expressed as Relative Light Unit (RLU) over the protein concentration. Graphs represent the mean normalized RLU +/− SD of triplicate values and are representative of 4 independent experiments for each cell line.

**Figure 7 viruses-15-02441-f007:**
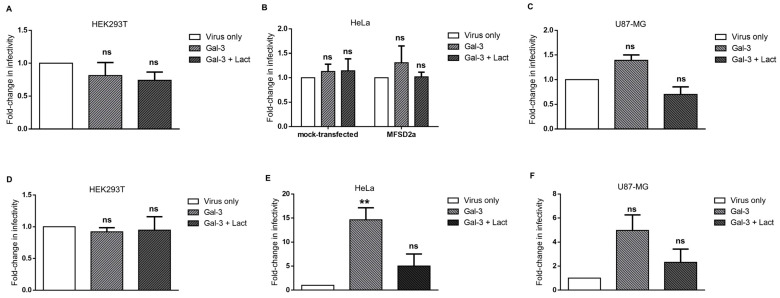
Different modulations of the infection of Syn-1- and Syn-2-pseudotyped viruses by Gal-3. HEK293T (**A**,**D**), HeLa (**B**,**E**), and U87-MG (**C**,**F**) cells were infected by Syn-2- (**A**–**C**) and Syn-1-pseudotyped (**D**–**F**) viruses in the presence of PBS (virus only), 4 µM Gal-3 (Gal-3), or 4 µM Gal-3 + 50 mM lactose (Gal-3 + Lact). In (**B**), HeLa cells were transiently transfected with an empty vector (mock-transfected) or with a MFSD2a expression vector prior to infection. Cells were lysed 24 h p.i., and luciferase activities were measured and normalized against protein concentration. Results are expressed as fold-change in infectivity compared to the control (virus only). Graphs represent the average fold-change in infectivity +/− SEM of three independent infection experiments for each cell line. ** *p* < 0.01; ns: non-significant.

## Data Availability

The data presented in this study are available on request from the corresponding author.

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
