# Peer review of "Galectin-1 Modulates the Fusogenic Activity of Placental Endogenous Retroviral Envelopes"

_viruses, 2023, doi:10.3390/v15122441_

Round 1

Reviewer 1 Report

Comments and Suggestions for Authors

The manuscripts describes the impact of GAL1 on fusogenic activity of selected endogenous retrovirus envelope proteins. Overall, the manuscript is well written and the presentation of the results is clear. The following points should be amended:

1.       Some of the results are presented relative to control values that have been set as 1 (fig 3, fig 4, fig. 5). This makes the interpretation difficult. Moreover the statistical analysis is not optimal because the control has no variance. The authors should present the raw data at least as supplementary data.

2.       The authors mention that the observed effects are partially cell type dependent. This cannot be concluded from the results because the authors used only one cell line per cell type (e.g one glioma cell line, one leukemia cell line, etc.). The authors should discuss the cell LINE specificity of the observations more carefully. Moreover, I suggest that the  authors include more discussion about the possible mechanisms that are responsible for the observed cell lines specificity.

Author Response

1.Some of the results are presented relative to control values that have been set as 1 (fig 3, fig 4, fig. 5). This makes the interpretation difficult. Moreover the statistical analysis is not optimal because the control has no variance. The authors should present the raw data at least as supplementary data.

We agree with the reviewers. In figure 3, 4, 5, fold values were calculated independently to control cells, that were set to a value of 1. To address the concern, and, as suggested, we provide the raw data as supplementary data 1for all three figures in the new version of the manuscripts and refer to the data in the text.

2. The authors mention that the observed effects are partially cell type dependent. This cannot be concluded from the results because the authors used only one cell line per cell type (e.g one glioma cell line, one leukemia cell line, etc.). The authors should discuss the cell LINE specificity of the observations more carefully. Moreover, I suggest that the  authors include more discussion about the possible mechanisms that are responsible for the observed cell lines specificity

The reviewer raises an important point and we have modified the text accordingly being more cautious. In the discussion section, we are also elaborating on the potential reasons for difference in cell specificity of action of galectins.

Reviewer 2 Report

Comments and Suggestions for Authors

In the study "Galectin-1 modulates the fusogenic activity of placental endogenous retroviral envelopes” Toudic and coauthors have compared the impact of Gal-1 on ERV envelope proteins (syn1 and 2 and EnvP(b)) and the infectivity of syncytins.

The study is well constructed and is novel in design and research methods. The results provide information on the fusogenic activity of three syn and the functional associations between the three proteins env and gal-1, which are useful for understanding their role in placental formation and function and in other biological and pathological processes. The results are interestingly presented and well discussed in the context of the available literature.

Author Response

No responses were needed.